# Serotonin in the Pathogenesis of Lymphocytic Colitis

**DOI:** 10.3390/jcm10020285

**Published:** 2021-01-14

**Authors:** Cezary Chojnacki, Tomasz Popławski, Anita Gasiorowska, Jan Chojnacki, Janusz Blasiak

**Affiliations:** 1Department of Clinical Nutrition and Gastroenterological Diagnostics, Medical University of Lodz, 90-647 Lodz, Poland; jan.chojnacki@umed.lodz.pl; 2Department of Molecular Genetics, Faculty of Biology and Environmental Protection, University of Lodz, 90-236 Lodz, Poland; tomasz.poplawski@biol.uni.lodz.pl (T.P.); janusz.blasiak@biol.uni.lodz.pl (J.B.); 3Department of Gastroenterology, Medical University of Lodz, 90-647 Lodz, Poland; anita.gasiorowska@umed.lodz.pl

**Keywords:** lymphocytic colitis, enteroendocrine cells, tryptophan hydroxylase, serotonin, 5-hydroxyindoleacetic acid, budesonide

## Abstract

Lymphocytic colitis (LC) is a chronic inflammatory disease associated with watery diarrhea, abdominal pain, and colonic intraepithelial lymphocytosis. Serotonin (5-hydroxytryptamine, 5-HT) is reported to increase in certain colon diseases; however, little is known regarding its metabolism in LC. In the present work, the level of 5-HT in serum and the number of enteroendocrine cells (EECs) as well as the expression of the 5-HT rate-limiting enzyme tryptophan hydroxylase 1 (TPH1) in colonic biopsies and urine 5-hydroxyindoeoacetic acid (5-HIAA) were determined in 36 LC patients that were treated with budesonide and 32 healthy controls. The 5-HT serum and 5-HIAA urine levels were measured using ELISA, the EEC number was determined immunohistochemically, and the colonic TPH1 mRNA expression was determined using RT-PCR. The levels of 5-HT and 5-HIAA and the number of EECs were higher in LC patients than in the controls, and positive correlations were observed between the 5-HT and 5-HIAA levels, 5-HT and EEC number, TPH1 mRNA and EEC number, as well as the severity of disease symptoms and 5-HIAA. Budesonide decreased the levels of 5-HT, 5-HIAA, and TPH1 expression and the number of EECs to values that did not differ from those for controls. In conclusion, the serotonin metabolism may be important for LC pathogenesis, and the urinary level of 5-HIAA may be considered as a non-invasive marker of this disease activity.

## 1. Introduction

Lymphocytic colitis (LC), a major subtype of microscopic colitis, is characterized by chronic non-bloody diarrhea, cramping abdominal pain, and other gastrointestinal complaints (reviewed in [1]). The LC incidence ranges from 2.3 to 16 per 100,000 and is higher in Northern Europe and northern states of the US [2,3]. The pathogenesis of LC, although not clear, is likely multifactorial and involves the immune response to luminal factors, such as bacteria and exogenous toxins as putative triggers, genetic factors, dysfunction of the epithelial barrier, dysfunction in the colonic neuroendocrine system, and several environmental and lifestyle risk factors, including female gender, smoking, drug side effects, and autoimmune diseases (reviewed in [4]).

The main histological finding in patients with LC is the presence of at least 20 intraepithelial lymphocytes per 100 epithelial cells [5]. The determination of precise number of intraepithelial lymphocytes may be assisted by cluster of differentiation 8 (CD8) and CD3 staining as the lymphocytes belong to the CD8+ cells bearing α/β T-cell receptors [6]. Intraepithelial lymphocytosis can be also seen within the crypts and the surface epithelium [7]. Colonoscopy typically shows a normal colonic mucosa; however, nonspecific alterations, including hyperemia, edema, loss of vascular pattern, and patchy erythema may be observed [8]. Apart from intraepithelial lymphocytes, plasma cells, eosinophils, mast cells, macrophages, and neutrophils are also present in the lamina propria [9].

In our previous work, we observed a higher number of colonic enterochromaffin (EC) cells, a subpopulation of enteroendocrine cells (EECs), in patients with LC compared to healthy subjects [10]. Similar results were obtained in other clinics and laboratories [11,12]. Most of the EC cells were in the glandular epithelium of the colonic mucosa. Enterochromaffin cells are the main source of serotonin (5-hydroxytryptamine, 5-HT), of which the majority in the human body is produced and stored in the gastrointestinal tract [13]. Therefore, the results showing an increase in EC cells in LC patients inspired us to ask whether that increase was associated with increased 5-HT production and secretion.

This question is further justified by the fact that 5-HT released from EC cells mediates many functions of the gastrointestinal tract, such as secretion, peristalsis, vasodilation, and the perception of pain, and nausea [14]. This is effectuated by the activation of a disparate family of 5-HT receptors on intrinsic and extrinsic afferent nerve fibers. Another stimulation to investigate the levels of 5-HT in patients with LC is the observation that patients with ulcerative colitis showed a reduced mucosal 5-HT content and a decreased EC cell population in severe ulcerative colitis samples [15].

Within the bowel, 5-HT is produced by EC cells and by serotonergic neurons of the myenteric plexus [11]. Both kinds of cells have rate-limiting enzymes in 5-HT biosynthesis, tryptophan hydroxylase-1 (TPH1, EC cells) and tryptophan hydroxylase-2 (TPH2, enteric and central serotonergic neurons) encoded by two different genes [16,17]. Upon release, 5-HT is inactivated by the serotonin reuptake transporter and broken down into 5-hydroxyindoleaceticacid (5-HIAA), which is excreted in the urine. Several studies show that 5-HIAA levels and HIAA/5-HT ratio may reflect an altered serotonin metabolism [18,19]. Enterochromaffin cells produce much more 5-HT than central and peripheral neurons [20]. Consequently, 5-HT derived from EC cells in the intestines overflows into the gut lumen and blood stream [19,20].

In the present work, we searched for the association between the severity of LC and 5-HT levels in the serum of LC patients, the number of EECs, and the TPH1 expression in the colonic mucosa, as well as the levels of 5-HIAA in the urine of these patients.

## 2. Materials and Methods

### 2.1. Patients

Thirty-six LC patients recruited from the Department of Gastroenterology, Medical University of Lodz, Lodz, Poland, in 2009–2017, were enrolled in this study. All patients represented newly diagnosed LC cases. All subjects underwent clinical, endoscopic, and histological examinations of the duodenal, small intestine, and colonic mucosa. Only patients without inflammatory changes throughout the large intestine were included in this study. The inclusion criteria were the presence of at least 20 lymphocytes per 100 colonic epithelial cells with enhanced inflammatory infiltrate in the lamina propria, but with no changes in the subepithelial collagen band.

The exclusion criteria were ulcerative colitis, Crohn’s disease, small intestinal bacterial overgrowth, and food intolerance, including celiac disease, exocrine pancreatic deficiency, thyroid dysfunction, metabolic and mental diseases, and a daily use of any drug in the last one month. All of the patients had intensive and chronic symptoms, such as watery, non-bloody diarrhea, abdominal pain, or stool incontinence. Colonoscopy showed a normal colonic mucus. The number of daily stools, nocturnal passage of stools, duration of loose stools, and the intensity of abdominal pain were determined as the severity index of the LC, using the 10-points Visual Analog Scale [21]. Thirty-two age-matched healthy subjects served as controls.

The study was conducted in accordance with the Declaration of Helsinki and the principles of Good Clinical Practice. Written consent was obtained from each subject enrolled in the study and the study protocol was approved by the Bioethics Committee of Medical University of Lodz (permit number RNN/242/06/KB dated 19 October 2006).

### 2.2. Laboratory Tests and Clinical Examinations

The following routine laboratory tests were performed in all subjects: blood cell count, quantification of protein, glucose, profile of lipids, bilirubin, iron, urea, creatinine, thyroid stimulating hormone, free thyroxine, free triiodothyronine antibodies to tissue transglutaminase, and deaminated gliadin peptide concentration, as well as the activity of alanine and asparagine aminotransferase, alkaline phosphatase, gamma-glutamyl transpeptidase, amylase and lipase.

The serum concentration of C-reactive protein (CRP) was determined by a latex agglutination photometric assay in COBAS INTEGRA 800 (Roche Diagnostic, Basel, Switzerland), the fecal calprotectin (FC) was evaluated by a sandwich ELISA test in Quantum Blue Reader (Buhlmann Diagnostics, Amherst, NH, USA), and the serum serotonin level and urine 5-HIAA concentrations were determined with ELISA.

Once diagnosed, the patients were instructed to administrate the same diet and take budesonide at a dose of 9.0 mg daily for three months in an open trial. Clinical follow-up was performed after 1, 2, and 3 months and endoscopic, histological and laboratory examinations were performed after 3 months.

On the day of the evaluation, all patients were administered the same liquid diet (Nutridrink, Nutricia, Hoofddorp, The Netherlands) in the amount of 3 × 400 mL, containing 18.9 g carbohydrate, 6.0 g protein, and 5.8 g lipid/mL, with a total caloric value of 1800 kcal and 1500 mL of isotonic water. Biopsy specimens were collected from the right, transverse and left colon. The number of the intraepithelial lymphocytes was determined by hematoxylin and eosin staining.

Venous blood and 24-hour urine collection samples were centrifuged and stored at −70 °C. The serum 5-HT and urinary 5-HIAA concentrations were measured by ELISA with the use of Immuno-Biological Laboratories kits no. RE 59121 and 59131 (Männedorf, Switzerland), respectively. The measurements were performed by photometry at the wavelength 450 nm in a microplate reader, model Expert 96-Reader (Biogenet, Jozefow, Poland). The obtained results of 5-HIAA were converted from nanograms per milliliter to micrograms per 24 h.

To determine the number of EECs, immunohistochemical staining was applied with mouse monoclonal antibodies (chromogranin A-LK 2H10, Cell Marque Co., Hague, The Netherlands) in the range of 10 fields in each biopsy specimen at 40× magnification. The number of the intraepithelial lymphocytes (IELs) and EECs was assessed with the UltraVision Quanto Detection System (Immunologic BV, Duiven, the Netherlands) computer program. The level of TPH1 mRNA expression was determined by RT-PCR with 50 mg of colonic tissues per individual. Briefly, colonic tissues were rapidly permeated to stabilize and protect cellular RNA with the RNA stabilization reagent RNAlater^®^ (Qiagen, Hilden, Germany). Prior to isolation of the total RNA, fragments of colonic tissues were homogenized with TissueRuptor (Qiagen). Then, the total RNA was isolated using a Qiagen RNeasy Plus Mini Kit (Qiagen) according to the manufacturer’s protocol.

The quantity and quality of the isolated RNA were estimated spectrophotometrically using a Take3 plate on a Synergy HT Microplate Reader (BioTek Instruments, Winooski, VT, USA). The real-time gene expression analysis was performed using the TaqMan Gene Expression Assays (Thermo Fisher Scientific, Waltham, MA, USA) with probes specific for TPH1 and SensiFAST TM Probe No-ROX One-Step Kit (Bioline, Taunton, MA, USA). The hypoxanthine phosphoribosyltransferase (HPRT, Assay ID: Hs01003267_m1) gene was used as a reference. Real-time PCR was performed with Bio-Rad CFX96 (Bio-Rad, Hercules, CA, USA). Expression analysis was performed with CFX Manager 1.6 software (Bio-Rad) with the ΔΔCt method [22].

### 2.3. Data Analysis

The Shapiro–Wilk W-test was used to test whether the data obeyed the normal distribution. The differences between two groups were determined using Student’s t-test or the Mann–Whitney U-test. The Wilcoxon matched pairs signed rank test was used to assess the difference between each set of matched pairs before and after treatment. Correlations between the quantitative variables were analyzed using Pearson’s correlation coefficient when the distributions of them were normal, otherwise Spearman’s correlation coefficient was used.

## 3. Results

The general characteristics of the individuals enrolled in this study and the results of the routine laboratory tests are presented in Table 1.

The serum concentration of 5-HT in LC patients was higher than in the controls (*p* < 0.05). The number of EECs and IELs, the expression of the *TPH1* gene, and the urinary excretion of 5-HIAA were higher in LC patients than in controls (*p* < 0.001). Positive correlations between the number of EECs and TPH1 expression (*p* < 0.05, Figure 1) and the serum 5-HT level (*p* < 0.05, Figure 1) were observed.

A positive correlation was found between the serum serotonin concentration and urinary 5-hydroxyindoleacetic acid (5-HIAA) excretion (*p* < 0.05, Figure 2).

A positive correlation was observed between the urinary 5-HIAA urine excretion and intensity of symptoms (*p* < 0.05, Figure 3).

Budesonide was well tolerated, and, after three months, the diarrhea and abdominal pain were completely resolved in 30 patients, and the intensity of the other symptoms apparently decreased in a further six patients. Improvement was also achieved in the results of the laboratory tests (Table 2).

Thirty-four patients consented to follow-up endoscopy with biopsy and biochemical examinations. In this group, the number of IELs and EECs in the colonic mucosa also decreased. Figure 4 presents the values of the biochemical parameters in LC patients who underwent budesonide therapy related to the corresponding values in healthy controls to facilitate comparison of the efficacy of the treatment.

## 4. Discussion

Microscopic colitis is a serious gastrointestinal problem as its prevalence in some populations is similar to Crohn’s disease and ulcerative colitis [23]. LC causes high overall symptom burdens and severely affects the quality of life. Despite this, its pathogenesis is poorly known, which likely hampers progress in effective treatment methods. Therefore, evidence-based treatment should be accompanied by experimental studies. Our results show an increased serum level of 5-HT in LC patients as compared with the healthy controls.

The 5-HT influences the immune response, and, thus, it can influence intestinal inflammation [24]. Increased levels of 5-HT were shown in patients with Crohn’s disease and ulcerative colitis and were shown to correlate with an increase in EC cell numbers [25]. Kwon et al. showed that serotonin signaling might influence the intestinal immune response through the modulation of the gut microbiota composition and resulted in increased susceptibility to colitis [26]. Although enhanced levels of 5-HT were reported in several studies in various colon inflammatory conditions, research in LC patients is scarce [10,15]. Fernández-Banares et al. showed that the use of a selective serotonin reuptake inhibitor (SSRI) was associated with the development of microscopic colitis, both collagenous and lymphocytic [27]. SSRIs increase the accessibility of 5-HT in the gut; therefore, the increased levels of 5-HT may play a role in LC pathogenesis. However, these results cannot be generalized as the cause-effect relationship for drug exposure and microscopic colitis can be considered only for certain drugs and in specific individual cases [28].

A treatment with duloxetine, a serotonin-norepinephrine reuptake inhibitor (SNRI), resulted in the development of LC, which was remitted after the drug withdrawal [29]. El-Salhy et al. observed a pronounced increase in serotonin cells in the colon epithelium of patients with LC [12]. They concluded that this increase resulted from the interaction of serotonin cells with immune cells.

The primary goals in LC treatment are to reach clinical remission, to improve the patient’s quality of life, and to follow evidence-based recommendations [1]. Oral administration with budesonide is the only drug therapy established as an effective treatment in microscopic colitis in randomized placebo-controlled trials [30]. Mesalazine and mesalazine plus cholestyramine were also tested [30]. We confirmed the efficacy of budesonide in LC patients—the drug effectively abolished two major LC symptoms: diarrhea and abdominal pain. Budesonide treatment also improved biochemical features associated with LC—it decreased the number of EECs and IELs as well as the 5-HT, 5-HIAA, FC, and CRP levels. However, the average number of lymphocytes infiltrating epithelial cells (>26) was still above the criterion of LC (at least 20). Therefore, this feature remains after LC symptoms have relieved, so its suitability as an early LC marker requires further research.

This appears to be especially important in LC as it is a disease with few specific clinical characteristics, and new markers of it are still needed. Various antibodies were found in the serum of LC patients; however, their profile is not very helpful in the evaluation of LC [31]. Research showed that LC patients had increased tumor necrosis factor alpha (TNF-α), interferon gamma (INF-γ), and interleukin 8 (IL-8) along with an upregulation of prostaglandin E receptor 4 (EP4); however, in general, proinflammatory cytokines have not been recognized as an independent LC marker [32,33]. Promising results were obtained in a study on the proteins produced by neuroendocrine cells, such as chromogranin or secretoneurin [11]. However, the LC diagnosis and, in fact, the LC definition are based on the number of lymphocytes penetrating the colon epithelium—at least 20 per 100 epithelial cells. However, clinical symptoms of LC also occur with fewer infiltrating lymphocytes (incomplete LC) [1].

We observed an increased expression of TPH1 in patients with LC, and this confirmed our previous observations [34]. This is not surprising in the context of increased 5-HT production, as TPH1 is a rate-limiting enzyme in 5-HT biosynthesis in EC cells. In this work, we observed an increase in the number of EECs; however, the relationship between the TPH1 expression and the number of EECs was not directly proportional (Figure 1). We measured the total level of 5-HT resulting from EECs and neurons. Therefore, the mechanisms regulating the expression of TPH1 in EECs might be changed due to the LC occurrence, but this assumption requires further research to precisely determine the relationship between the TPH1 expression and the number of EECs, both in normal and disease conditions. Inhibitors of TPH1 were shown to exert beneficial effects in patients with colitis [35].

The level of 5-HIAA in the urine reflects the amount of 5-HT that is inactivated and secreted, and, thus, it can be linked with the amount of active 5-HT, although, again, a precise relationship between the amount of active 5-HT and urine 5-HIAA should be established in further studies. In this work, we showed increased levels of 5-HIAA in LC patients, which is not very surprising as we also observed increased levels of 5-HT in these patients. We observed a positive correlation between the 5-HIAA levels and the severity of LC symptoms (Figure 3). Decreased 5-HIAA levels associated with the use of a TPH1 inhibitor were reported to correlate with the improvement of irritable bowel syndrome symptoms [36].

The results of many studies, including the present work, support the thesis on the importance of enteroendocrine cell-derived 5-HT in gut functions and especially gastrointestinal transit; however, recent data underline the emerging role of neuronal 5-HT in gastrointestinal processes [33]. Therefore, future research on the role of 5-HT in LC should address 5-HT from these two sources: enteroendocrine and neural.

Our study has several limitations that point at important elements of further research. First, the number of subjects enrolled was not very high; however, they were a relatively homogenous and well-diagnosed group. Secondly, to assess disease symptoms, only the Visual Analogue Scale was used, as there is still a lack of established criteria for assessing activity of lymphocytic colitis. We established the LC diagnosis based on of the number of intraepithelial lymphocytes. However, an increased number of these cells can be seen in other inflammatory conditions in the colon, including inflammatory bowel disease (IBD). The analysis of CD3+ lymphocyte collection in the mucosal/epithelial layer would increase the specificity of the diagnosis [33]. We used mouse monoclonal antibodies (chromogranin A), which were appropriate to identify enteroendocrine cells in the gut; however, to specifically detect ES cells, an additional antibody against 5-HT should be used.

We looked for a correlation between the 5-HT levels and the number of EECs. However, 5-HT can be stored in murine mast cells and some authors reported that human mast cells contained 5-HT as well [37]. Although, at present, the secretion of 5-HT from such cells is not known, this possibility cannot be excluded. We showed a positive correlation between the 5-HT levels and the number of EECs. Further studies on the potential significance of 5-HT from sources other than EECs are needed. We used the Visual Analog Scale and not the Hjortswang Criteria or microscopic colitis disease activity index (MCDAI) as it is more universal and not biased towards collagenous colitis [38].

In conclusion, lymphocytic colitis can be associated with increased levels of 5-HT, which may result from an increased number of EC cells within increased EECs population and an increased expression of TPH1, a rate-limiting enzyme in 5-HT production. The urinary level of 5-HIAA, the final product of 5-HT breakdown, correlated with the TPH1 expression and the severity of LC, and thus it can be used as a non-invasive marker of the activity of this disease. The administration of budesonide, a first-line treatment for active LC, decreased the 5-HT levels, TPH1 expression, and EECs number in LC patients. Altogether, the 5-HT metabolism may play a role in LC pathogenesis, and the urinary 5-HIAA level may be considered as a non-invasive indicator of this disease activity.

## Figures and Tables

**Figure 1 jcm-10-00285-f001:**
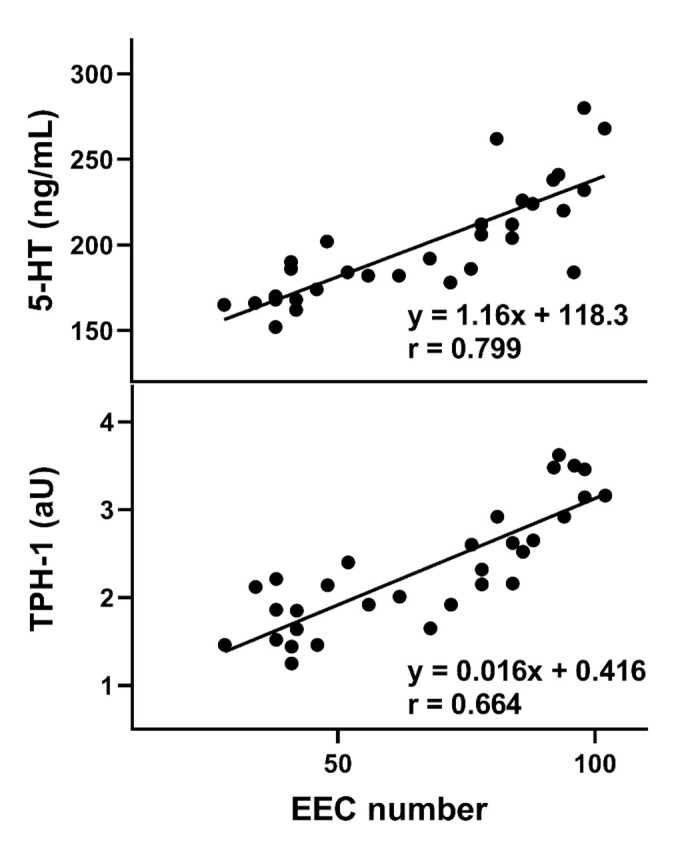
Correlation between number of colonic enteroendocrine (EEC) cells and serum serotonin (5-HT) levels (upper panel) and tryptophan hydroxylase-1 (TPH1) mRNA expression levels (lower panel) in lymphocytic colonic patients (*n* = 36). The Spearman rank correlation coefficient (r) was used to evaluate of the strength of correlation. The regression line was drawn using the least squares method.

**Figure 2 jcm-10-00285-f002:**
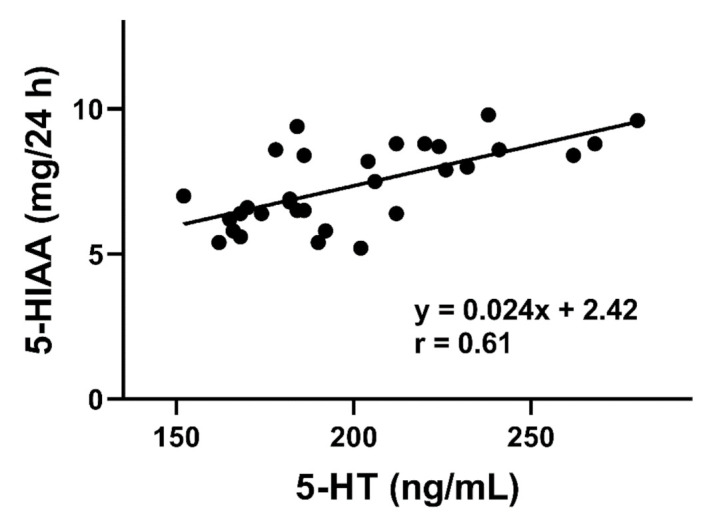
Correlation between the serum serotonin (5-HT) levels and urinary 5-hydroxyindoleaceticacid (5-HIAA) concentrations expressed in mg released to the urine in 24 h in lymphocytic colitis patients (*n* = 36). The Pearson rank correlation coefficient (r) was used to evaluate of the strength of correlation. The regression line was drawn using the least squares method.

**Figure 3 jcm-10-00285-f003:**
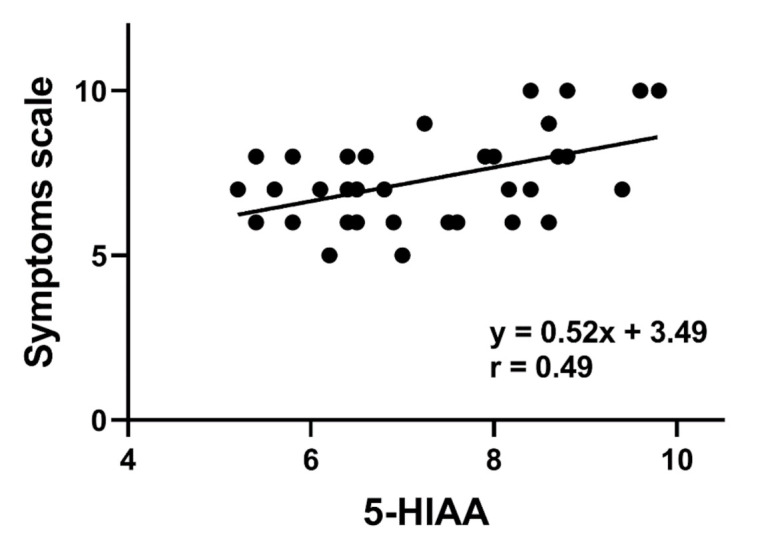
Correlation between urinary 5-hydroxyindoleaceticacid (5-HIAA) concentrations expressed in ng per urine mL and the intensity of symptoms assessed by the Visual Analog Scale in lymphocytic colitis patients. The Spearman rank correlation coefficient (r) was used to evaluate of the strength of correlation. The regression line was drawn using the least squares method.

**Figure 4 jcm-10-00285-f004:**
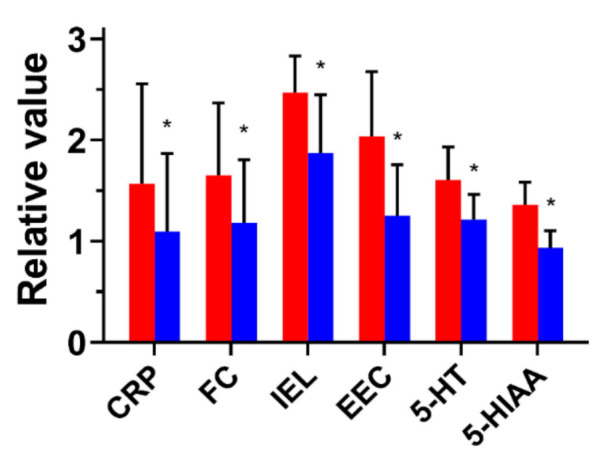
The biochemical parameters in lymphocytic colitis patients treated for 3 months with budesonide. The mean values of the parameters normalized to values for healthy controls are presented before (red bars) and after (blue bars) treatment. CRP, C-reactive protein; FC, fecal calprotectin; IEL, intraepithelial lymphocytes (per 100 enterocytes); EEC, enterochromaffin cells in the range 10 fields in each biopsy specimen; 5-HT, serum serotonin; 5-HIAA, urinary 5-hydroxyindoleacetic acid; Mean ± SD; *, *p* < 0.05 as compared with the values before treatment.

**Table 1 jcm-10-00285-t001:** Characteristics of lymphocytic colitis (LC) patients and control individuals enrolled in this study.

Feature ^a^	Controls	LC Patients
Number	32	36
Gender		
M	16	16
F	16	20
CRP (mg/L)	1.97 ± 1.05	2.96 ± 1.78
FC (µg/g)	26.4 ± 9.66	44.2 ± 21.9
IEL (number)	14.2 ± 3.42	34.1 ± 5.83 ***
EEC (number)	33.5 ± 9.78	67.9 ± 21.8 ***
TPH1 (aU)	1.18 ± 0.40	2.27 ± 0.67 ***
5-HT (ng/mL)	125.5 ± 34.6	203.9 ± 42.6 **
5-HIAA (mg/24 h)	5.52 ± 1.24	7.46 ± 1.25 ***

^a^ average ± SD; SD, standard deviation; M, male; F, female; CRP, C-reactive protein; FC, fecal calprotectin; IELs, intraepithelial lymphocytes (per 100 enterocytes); EECs, enteroendocrine cells in the range 10 field in each biopsy specimen; TPH1, tryptophan hydroxylase; 5-HT, serum serotonin; 5-HIAA, urinary 5-hydroxyindoleacetic acid; **, *p* < 0.01; and ***, *p* < 0.001 as compared with healthy controls.

**Table 2 jcm-10-00285-t002:** Histological and biochemical features of 34 patients with lymphocytic colitis (LC) before and after a three-month treatment with budesonide.

Feature ^a^	Before	After
CRP (mg/L)	3.09 ± 1.95	2.16 ± 1.52 *
FC (µg/g)	43.6 ± 18.9	31.3 ± 16.4 *
IEL (number)	35.1 ± 5.12	26.6 ± 8.20 *
EEC (number)	68.2 ± 21.5	42.0 ± 16.9 *
5-HT (ng/mL)	201.5 ± 41.3	152.4 ± 31.3 *
5-HIAA (mg/24 h)	7.52 ± 1.23	5.16 ± 0.93 *

^a^ average ± SD; SD, standard deviation; CRP, C-reactive protein; FC, fecal calprotectin; IEL, intraepithelial lymphocytes (per 100 enterocytes); EEC, enteroendocrine cells in the range of 10 fields in each biopsy specimen; TPH1, tryptophan hydroxylase; 5-HT, serum serotonin; 5-HIAA, urinary 5-hydroxyindoleacetic acid; *, *p* < 0.05 as compared with patients before treatment.

## Data Availability

The data that support the findings of this study are available on request from the corresponding author. The data are not publicly available due to privacy and ethical restrictions and are stored at www.umed.lodz.pl.

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
