# Peer review of "Serotonin in the Pathogenesis of Lymphocytic Colitis"

_jcm, 2021, doi:10.3390/jcm10020285_

Round 1

Reviewer 1 Report

Introduction: Northern states of of the US should just be northern states of the US.

Materials and Methods:  what is meant by permanent use of any drug in the last month?

Line 224:  sentence does not make sense

Author Response

Comment (C): Introduction: Northern states of of the US should just be northern states of the US.

Answer (A): We have corrected that.

C: Materials and Methods:  what is meant by permanent use of any drug in the last month?

A: We have changed into “daily use”

C: Line 224:  sentence does not make sense

A: We have changed that non-completed sentence into:

“Moreover, budesonide treatment also improved biochemical features associated with LC – decreased number of EC cells and IEL as well as 5-HT, 5-HIAA, FC and CRP levels.”

Reviewer 2 Report

The authors have undertaken an interesting and clinically relevant study to investigate the potential relationship between serotonin (5-HT) signaling and lymphocytic colitis (LC) along with the impact of steroid therapy on 5-HT-associated factors.  Previous studies have evaluated 5-HT signaling elements in the context of several gastrointestinal conditions, including inflammatory bowel diseases, irritable bowel syndrome, diverticular disease and microscopic colitides (including LC).  A few previous studies have demonstrated an increase in enteroendocrine cells (El-Salhy et al., Dig Dis Sci 2012) and enterochromaffin cells in LC (El-Salhy et al., WJG 2012).  However, no previous study has attempted to perform a more complete evaluation of 5-HT signaling in the setting of LC and that is what this study was designed to do.  To be clear, this does not represent a comprehensive evaluation of 5-HT signaling as there are several important components missing, including assessments of total intestinal 5-HT content, 5-HT receptor expression, 5-HT uptake capacity (i.e., SERT expression/function) and metabolism.  This study is also unique, though, in its prospective assessment of the effects of budesonide on 5-HT signaling elements in LC.  There are some potential methodological issues that need to be addressed (described below) but, nonetheless, this study represents an interesting follow-up investigation to the authors’ previous study, and (if the issues noted below can be adequately addressed) could represent a step forward in our understanding of the potential role that 5-HT plays in LC. In regard to the manuscript, in addition to some methodological questions, there are scattered issues with syntax and grammar.  If all of these addresses can be appropriately addressed, though, I think this paper should be considered for publication.

  • Introduction, page 1, paragraph 1, line 32 -> Remove “a” before “chronic non-bloody diarrhea”.
  • Introduction, page 1-2, paragraph 1, line 44+ -> Include reference to other studies that have demonstrated increased numbers of enterochromaffin cells in LC (El-Salhy et al., Dig Dis Sci 2012, 57(12):3154-9; El-Salhy et al., WJG 2012, 18(42): 6070–6075).
  • Introduction, page 2, paragraph 2, lines 63-64 -> Change “In consequence, EC-cells derived 5-HT overflows to achieve the gastrointestinal lumen and blood.”” to “As a consequence, 5-HT derived from EC cells in the intestines overflows into the gut lumen and blood stream.”
  • Methods, page 2, Patients -> The inclusion criteria are a bit confusing. Were all of the study participants new diagnoses of LC?  If so, that should be clearly stated.  Additionally, I understand using a specific cut-off for lymphocyte density as part of the identification of the study cohort but there was no mention of ensuring a formal diagnosis of LC.  This is important because other GI conditions are associated with lymphocytic infiltrate in the colon (ex: IBD). Even when indicating that patients with other inflammatory conditions (such as IBD) are excluded, it is possible these histological changes could represent early stage IBD or other diagnoses.  Using patients that have a clearly established diagnosis helps make that possibility less likely.
  • Methods, page 3, Laboratory tests and clinical examinations -> The antibody described (mouse monoclonal ab against chromogranin A) is appropriate for identifying enteroendocrine cells in the gut but an additional antibody (against 5-HT) is required to specifically identify EC cells. Was this done?  If not, that represents a significant methodological limitation.
  • Methods/Discussion -> Did the authors consider evaluating for mast cells in biopsy samples? These could also serve as an important source of 5-HT from the gut. If not, it would be helpful to talk about this as a potential confounder in the limitations section within the Discussion.
  • Discussion -> There are several editing mistakes sprinkled throughout this section, including missing articles (ex: the) and words that should pluralized. Please work with an editor to identify and address them appropriately.

Author Response

Comments and Suggestions for Authors

The authors have undertaken an interesting and clinically relevant study to investigate the potential relationship between serotonin (5-HT) signaling and lymphocytic colitis (LC) along with the impact of steroid therapy on 5-HT-associated factors.  Previous studies have evaluated 5-HT signaling elements in the context of several gastrointestinal conditions, including inflammatory bowel diseases, irritable bowel syndrome, diverticular disease and microscopic colitides (including LC).  A few previous studies have demonstrated an increase in enteroendocrine cells (El-Salhy et al., Dig Dis Sci 2012) and enterochromaffin cells in LC (El-Salhy et al., WJG 2012).  However, no previous study has attempted to perform a more complete evaluation of 5-HT signaling in the setting of LC and that is what this study was designed to do.  To be clear, this does not represent a comprehensive evaluation of 5-HT signaling as there are several important components missing, including assessments of total intestinal 5-HT content, 5-HT receptor expression, 5-HT uptake capacity (i.e., SERT expression/function) and metabolism.  This study is also unique, though, in its prospective assessment of the effects of budesonide on 5-HT signaling elements in LC.  There are some potential methodological issues that need to be addressed (described below) but, nonetheless, this study represents an interesting follow-up investigation to the authors’ previous study, and (if the issues noted below can be adequately addressed) could represent a step forward in our understanding of the potential role that 5-HT plays in LC. In regard to the manuscript, in addition to some methodological questions, there are scattered issues with syntax and grammar.  If all of these addresses can be appropriately addressed, though, I think this paper should be considered for publication.

Comment (C): Introduction, page 1, paragraph 1, line 32 -> Remove “a” before “chronic non-bloody diarrhea”.

Answer (A): We have removed that.

C: Introduction, page 1-2, paragraph 1, line 44+ -> Include reference to other studies that have demonstrated increased numbers of enterochromaffin cells in LC (El-Salhy et al., Dig Dis Sci 2012, 57(12):3154-9; El-Salhy et al., WJG 2012, 18(42): 6070–6075).

A: We have added the following sentence to the Introduction section (lines 46-47 in the original submission):

“Similar results were obtained in other clinics and laboratories [11,12] “

with the following new references:

  1. El-Salhy, M.; Gundersen, D.; Hatlebakk, J.G.; Hausken, T. Chromogranin A cell density as a diagnostic marker for lymphocytic colitis. Dig Dis Sci 2012, 57, 3154-3159, doi:10.1007/s10620-012-2249-6.
  2. El-Salhy, M.; Gundersen, D.; Hatlebakk, J.G.; Hausken, T. High densities of serotonin and peptide YY cells in the colon of patients with lymphocytic colitis. World journal of gastroenterology 2012, 18, 6070-6075, doi:10.3748/wjg.v18.i42.6070..

C: Introduction, page 2, paragraph 2, lines 63-64 -> Change “In consequence, EC-cells derived 5-HT overflows to achieve the gastrointestinal lumen and blood.”” to “As a consequence, 5-HT derived from EC cells in the intestines overflows into the gut lumen and blood stream.”

A: We have done so.

C: Methods, page 2, Patients -> The inclusion criteria are a bit confusing. Were all of the study participants new diagnoses of LC?  If so, that should be clearly stated. 

A: We have changed 2.1. Patients section into:

“Thirty-six LC patients recruited from the Department of Gastroenterology, Medical University of Lodz, Lodz, Poland in 2009-2017 were enrolled in this study. All patients represented newly diagnosed LC cases. All subjects underwent clinical, endoscopic and histological examinations of duodenal, small intestine, and colonic mucosa. Only patients without inflammatory changes throughout the large intestine were included in this study. The inclusion criteria were the presence of at least 20 lymphocytes per 100 colonic epithelial cells with enhanced inflammatory infiltrate in the lamina propria, but with no changes in the subepithelial collagen band. The exclusion criteria were ulcerative colitis, Crohn disease, small intestinal bacterial overgrowth, food intolerance, including celiac disease exocrine pancreatic deficiency, thyroid dysfunction, metabolic and mental diseases and a daily use of any drug in the last one month. All the patients had intensive and chronic symptoms such as watery, non-bloody diarrhea, abdominal pain, stool incontinence. Colonoscopy showed a normal colonic mucus.

The number of daily stools, nocturnal passage of stools, duration of loose stools, and intensity of abdominal pain were determined as severity index of LC, using the 10-points Visual Analog Scale [21]. Thirty-two age-matched healthy subjects served as controls.

The study was conducted in accordance with the Declaration of Helsinki and the principles of Good Clinical Practice. Written consent was obtained from each subject enrolled in the study and the study protocol was approved by the Bioethics Committee of Medical University of Lodz (permit number RNN/242/06/KB dated October 19, 2006).”

C: Additionally, I understand using a specific cut-off for lymphocyte density as part of the identification of the study cohort but there was no mention of ensuring a formal diagnosis of LC. This is important because other GI conditions are associated with lymphocytic infiltrate in the colon (ex: IBD). Even when indicating that patients with other inflammatory conditions (such as IBD) are excluded, it is possible these histological changes could represent early stage IBD or other diagnoses.  Using patients that have a clearly established diagnosis helps make that possibility less likely.

A: We have added the following fragment to the Discussion section in the description of the limitations of our study:

“We established the LC diagnosis on the base of the number intraepithelial lymphocytes. However, an increased number of these cells can be seen in other inflammatory conditions in the colon, including IBD. The analysis of CD3+ lymphocyte collection in the mucosal/epithelial layer would increase the specificity of the diagnosis [40].”

with the new reference:

  1. Pisani, L.F.; Tontini, G.E.; Marinoni, B.; Villanacci, V.; Bruni, B.; Vecchi, M.; Pastorelli, L. Biomarkers and Microscopic Colitis: An Unmet Need in Clinical Practice. Front Med (Lausanne) 2017, 4, 54-54, doi:10.3389/fmed.2017.00054.

C: Methods, page 3, Laboratory tests and clinical examinations -> The antibody described (mouse monoclonal ab against chromogranin A) is appropriate for identifying enteroendocrine cells in the gut but an additional antibody (against 5-HT) is required to specifically identify EC cells. Was this done?  If not, that represents a significant methodological limitation.

A: We have added the following fragment to the Discussion section:

“We used mouse monoclonal antibodies (chromogranin A), which were appropriate to identify enteroendocrine cells in the gut, but to specifically detect ES cells an additional antibody against 5-HT should be used.”

C: Methods/Discussion -> Did the authors consider evaluating for mast cells in biopsy samples? These could also serve as an important source of 5-HT from the gut. If not, it would be helpful to talk about this as a potential confounder in the limitations section within the Discussion.

A: We have added the following fragment to Discussion when writing about limitations of our study:

“We looked for the correlation between 5-HT levels and the number of EC cells. However, 5-HT can be stored in murine mast cells and some authors reported that human mast cells contained 5-HT as well [43]. Although at present it is not exactly known about the secretion of 5-HT from such cells, this possibility cannot be excluded. However, we showed a positive correlation between 5-HT levels and the number of EC cells, but we do not claim that this was a mutually reciprocal correlation, i.e., that the levels of 5-HT were determined solely by the number of EC cells. Further studies on the potential significance of 5-HT from other sources than EC cells are needed.”

with the new reference:

43 Kritas, S.K.; Saggini, A.; Cerulli, G.; Caraffa, A.; Antinolfi, P.; Pantalone, A.; Rosati, M.; Tei, M.; Speziali, A.; Saggini, R., et al. Relationship between serotonin and mast cells: inhibitory effect of anti-serotonin. Journal of biological regulators and homeostatic agents 2014, 28, 377-380.

C: Discussion -> There are several editing mistakes sprinkled throughout this section, including missing articles (ex: the) and words that should pluralized. Please work with an editor to identify and address them appropriately.

A: We have done our best to correct all errors and mistakes in the manuscript.

Reviewer 3 Report

General comments:

In the paper by Chojnacki et al. the authors investigate the pathogenesis of lymphocytic colitis and identify the urinary level of 5-HIAA as a non-invasive biomarker of disease activity. The incidence of microscopic colitis has increased during the last decades and have in some countries now surpassed the incidence of IBD emphasizing the increased burden and the need of non-invasive biomarkers. The subject is therefore considered highly relevant.

However I think the paper could be improved in some points. Most important it would be easier to read if more structured and the primary inclusion criteria with histopathological features should be more clearly described.

Abstract:

Line 21: Change LA to LC.

Line 27: I think the authors mean disease activity? I agree that 5-HIAA may be a possible marker of disease activity but you still need to have a primary diagnosis by histological examination.

Introduction:

Line 32: LC is not a variant of MC but one of the two major subtypes

Line 39-40 and 42-43: Please describe the histological features together and not mixed with colonoscopy. Add intraepithelial in line 40 and in lamina propria in line 43 and use a reference focusing on histology.

Line 45: Add “the” before glandular and change “area” to “epithelium”

Line 46: Use abbreviation for EC

Line 47: Do you mean .., “of which the majority…”?

Line 52: Maybe the word “effectuated” is more appropriate compared with “done”

Line 53: Maybe the word “investigate” is more appropriate compared with “check”

Line 63: Maybe the word “reach” is more appropriate compared with “achieve”

Materials and methods:

Section 2.1.

I think the papers would be improved by focusing on the clinical criteria in this section and remove all histology to the next section. As in the introduction this has been described both line 71-72 and 76-77 and 78-79. And both 20 and 25 IELs have been used as lower cut-off…

Line: 76-77: How have you assessed that the inflammatory changes was present throughout the bowel? I guess histology as colonoscopy was normal? Describe this in details. Where each biopsy assessed separately and changes seen in all or only by location?

Line 83-84: Delete. Is also repeated in results.

Section 2.2.

The section with line 112-117 should be before line 100-111 as this is more logic. Please, describe how you assessed the number of IELs and EC cells. Digitized slides? Computer count or manual count or eyeballing?

Line 114: Change “method” to “staining”

Integrate section 2.3. in 2.2.

Results:

Figure 3: Change “Hamiltonian” to “Visual Analog Scale” if I understand correct

Line 184: Change “5” to “4”

Discussion:

Why did the authors choose staining with Chromogranin A which is a general marker for neuroendocrine cells and not serotonin?

Why did the authors choose Visual Analog Scale and not Hjortswang criteria (although mostly used for collagenous colitis) or the MCDAI (microscopic colitis disease activity index)?

Line 199 and 222: Maybe “molecular” is not the most correct word to use in this context, please consider alternatives.

Line 221: Correct “trials”

Line 224: Missing words

Line 226: In this context it seems wrong to consider number of IELs as an early marker of LC as the patients in your study have been diagnosed histologically and treated with budesonide and the number of IELs remains increased. It would be more appropriate to conclude that the histological features remains after symptoms have relieved. This fact have also been demonstrated by several other authors.

Line 229: Change “were” to “have been” if you are referring to other studies

Line 233-35: Increased number of IELs alone is definitely not enough for the diagnosis – this has to be accompanied by inflammation in lamina propria. Furthermore correlation to clinical symptoms should be emphasized as histology is not specific. 

Line 237: Change “earlier” to “previous”

Line 238: Change “surprised” to “surprising”

Line 247: Maybe “secreted” is preferable to “broken”

Line 250: As line 238

Line 268, 269, 270: Change “LA” to “LC”

Line 270: Add “activity”, se comment in abstract section

Author Response

General comments:

In the paper by Chojnacki et al. the authors investigate the pathogenesis of lymphocytic colitis and identify the urinary level of 5-HIAA as a non-invasive biomarker of disease activity. The incidence of microscopic colitis has increased during the last decades and have in some countries now surpassed the incidence of IBD emphasizing the increased burden and the need of non-invasive biomarkers. The subject is therefore considered highly relevant.

Comment (C): However I think the paper could be improved in some points. Most important it would be easier to read if more structured and the primary inclusion criteria with histopathological features should be more clearly described.

Answer (A):  We have changed 2.1. Patients section into:

“Thirty-six LC patients recruited from the Department of Gastroenterology, Medical University of Lodz, Lodz, Poland in 2009-2017 were enrolled in this study. All patients represented newly diagnosed LC cases. All subjects underwent clinical, endoscopic and histological examinations of duodenal, small intestine, and colonic mucosa. Only patients without inflammatory changes throughout the large intestine were included in this study. The inclusion criteria were the presence of at least 20 lymphocytes per 100 colonic epithelial cells with enhanced inflammatory infiltrate in the lamina propria, but with no changes in the subepithelial collagen band. The exclusion criteria were ulcerative colitis, Crohn disease, small intestinal bacterial overgrowth, food intolerance, including celiac disease exocrine pancreatic deficiency, thyroid dysfunction, metabolic and mental diseases, and a daily use of any drug in the last one month. All the patients had intensive and chronic symptoms such as watery, non-bloody diarrhea, abdominal pain, stool incontinence. Colonoscopy showed a normal colonic mucus.

The number of daily stools, nocturnal passage of stools, duration of loose stools, and intensity of abdominal pain were determined as severity index of LC, using the 10-points Visual Analog Scale [21]. Thirty-two age-matched healthy subjects served as controls.

The study was conducted in accordance with the Declaration of Helsinki and the principles of Good Clinical Practice. Written consent was obtained from each subject enrolled in the study and the study protocol was approved by the Bioethics Committee of Medical University of Lodz (permit number RNN/242/06/KB dated October 19, 2006).”

C: Abstract: Line 21: Change LA to LC.

A: We have done so.

C: Line 27: I think the authors mean disease activity? I agree that 5-HIAA may be a possible marker of disease activity but you still need to have a primary diagnosis by histological examination.

A: We have deleted the word “independent” and added the word “activity” in the end of that sentence.

C: Introduction: Line 32: LC is not a variant of MC but one of the two major subtypes

A: We have changed “…a variant of microscopic colitis…” into: “…a major subtype of microscopic colitis…”

C: Line 39-40 and 42-43: Please describe the histological features together and not mixed with colonoscopy. Add intraepithelial in line 40 and in lamina propria in line 43 and use a reference focusing on histology.

A: We have changed the fragment:

“Histopathological criterion for LC is the presence of at least 20 lymphocytes per 100 colonic epithelial cells, usually without crypt distortion [5]. Colonoscopy usually shows a normal colonic mucosa, but nonspecific alterations, including hyperemia, edema, loss of vascular pattern, patchy erythema may be observed [6]. Apart from intraepithelial lymphocytes, plasma cells, eosinophils, mast cells, macrophages and neutrophils are also present [7].”

into:

“The main histological finding in patients with LC is the presence of at least 20 intraepithelial lymphocytes per 100 epithelial cells [5]. The determination of precise number of intraepithelial lymphocytes may be assisted by CD8 and CD3 staining as the lymphocytes belong to the CD8+ bearing α/β T-cell receptors [6]. Intraepithelial lymphocytosis can be also seen within the crypts and the surface epithelium [7]. Colonoscopy usually shows a normal colonic mucosa, but nonspecific alterations, including hyperemia, edema, loss of vascular pattern, patchy erythema may be observed [8]. Apart from intraepithelial lymphocytes, plasma cells, eosinophils, mast cells, macrophages and neutrophils are also present in lamina propria [9].”

with the following new references:

  1. Münch, A.; Aust, D.; Bohr, J.; Bonderup, O.; Fernández Bañares, F.; Hjortswang, H.; Madisch, A.; Munck, L.K.; Ström, M.; Tysk, C., et al. Microscopic colitis: Current status, present and future challenges: statements of the European Microscopic Colitis Group. Journal of Crohn's & colitis 2012, 6, 932-945, doi:10.1016/j.crohns.2012.05.014.
  2. Fiehn, A.M.; Engel, U.; Holck, S.; Munck, L.K.; Engel, P.J. CD3 immunohistochemical staining in diagnosis of lymphocytic colitis. Human pathology 2016, 48, 25-31, doi:10.1016/j.humpath.2015.09.037.
  3. Chetty, R.; Govender, D. Lymphocytic and collagenous colitis: an overview of so-called microscopic colitis. Nature reviews. Gastroenterology & hepatology 2012, 9, 209-218, doi:10.1038/nrgastro.2012.16.5.

C: Line 45: Add “the” before glandular and change “area” to “epithelium”

A: We have done

C: Line 46: Use abbreviation for EC

A: If not absolutely needed, we prefer not to use an abbreviation at the beginning of a sentence.

C: Line 47: Do you mean .., “of which the majority…”?

A: We have changed that.

C: Line 52: Maybe the word “effectuated” is more appropriate compared with “done”. Line 53: Maybe the word “investigate” is more appropriate compared with “check”. Line 63: Maybe the word “reach” is more appropriate compared with “achieve”

A: We have changed that.

C: Materials and methods: Section 2.1. I think the papers would be improved by focusing on the clinical criteria in this section and remove all histology to the next section. As in the introduction this has been described both line 71-72 and 76-77 and 78-79. And both 20 and 25 IELs have been used as lower cut-off…

A: This section has been changed as specified in the answer to the first comment.

C: Line: 76-77: How have you assessed that the inflammatory changes was present throughout the bowel? I guess histology as colonoscopy was normal? Describe this in details. Where each biopsy assessed separately and changes seen in all or only by location?

A: We are sorry – “with inflammatory changes” should write “without inflammatory changes”.

C: Line 83-84: Delete. Is also repeated in results.

A: We have done so.

C: Section 2.2. The section with line 112-117 should be before line 100-111 as this is more logic.

A: We have done so.

C: Please, describe how you assessed the number of IELs and EC cells. Digitized slides? Computer count or manual count or eyeballing?

A: We have changed the fragment:

“To determine the number of EC cells immunohistochemical method was applied with mouse monoclonal antibodies (chromogranin A – LK 2H10, Cell Marque Co., Hague, The Netherlands) and Ultravision Quanto Detection System (HRP-DAB, Immunologic BV, Duiven, The Netherlands) in the range 10 fields in each biopsy specimen at 40 × magnification.”

into:

“To determine the number of EC cells immunohistochemical staining was used with mouse monoclonal antibodies (chromogranin A – LK 2H10, Cell Marque Co., Hague, The Netherlands) in the range 10 fields in each biopsy specimen at 40 × magnification. The number of the IELs and EC cells was assessed with the Ultravision Quanto Detection System (HRP-DAB, Immunologic BV, Duiven, The Netherlands) computer program.”

C: Line 114: Change “method” to “staining”

A: We have done so.

C: Integrate section 2.3. in 2.2.

A: We have done so.

C: Results: Figure 3: Change “Hamiltonian” to “Visual Analog Scale” if I understand correct

A: We have done so.

C: Line 184: Change “5” to “4”

A: We have done so.

C: Discussion: Why did the authors choose staining with Chromogranin A which is a general marker for neuroendocrine cells and not serotonin?

A: We have added the following fragment to the Discussion section:

“We used mouse monoclonal antibodies (chromogranin A), which were appropriate to identify enteroendocrine cells in the gut, but to specifically detect ES cells an additional antibody against 5-HT should be used.”

C: Why did the authors choose Visual Analog Scale and not Hjortswang criteria (although mostly used for collagenous colitis) or the MCDAI (microscopic colitis disease activity index)?

A: We have added the following fragment to Discussion:

“We used Visual Analog Scale and not the Hjortswang Criteria or MCDAI (microscopic colitis disease activity index) as it is more universal and not biased towards collagenous colitis [44].”

with the following reference:

  1. Münch, A.; Bohr, J.; Miehlke, S.; Benoni, C.; Olesen, M.; Öst, Å.; Strandberg, L.; Hellström, P.M.; Hertervig, E.; Armerding, P., et al. Low-dose budesonide for maintenance of clinical remission in collagenous colitis: a randomised, placebo-controlled, 12-month trial. Gut 2016, 65, 47-56, doi:10.1136/gutjnl-2014-308363.

C: Line 199 and 222: Maybe “molecular” is not the most correct word to use in this context, please consider alternatives.

A: We have changed “molecular experimental studies” into “experimental studies” in line 199 and replaced ”molecular features” with “biochemical features” in lines 223-224 (original submission).

C: Line 221: Correct “trials”

A: We have corrected that.

C: Line 224: Missing words

A: We have replaced:

“Moreover, budesonide treatment also improved molecular features associated with LC – decreased number of EC cells and 5-HT levels as well as”

with:

“Moreover, budesonide treatment also improved biochemical features associated with LC – decreased number of EC cells and IEL as well as 5-HT, 5-HIAA, FC and CRP levels.”

C: Line 226: In this context it seems wrong to consider number of IELs as an early marker of LC as the patients in your study have been diagnosed histologically and treated with budesonide and the number of IELs remains increased. It would be more appropriate to conclude that the histological features remains after symptoms have relieved. This fact have also been demonstrated by several other authors.

A: We have changed the sentence:

“Therefore, this feature could be considered as an early marker of LC, but its specificity and sensitivity should be determined in further research.”

into:

“Therefore, this feature remains after LC symptoms have relieved and so its suitability as an early LC marker requires further research.”

C: Line 229: Change “were” to “have been” if you are referring to other studies

A: We consequently use the simple past tense throughout the manuscript to relate to other studies.

C: Line 233-35: Increased number of IELs alone is definitely not enough for the diagnosis – this has to be accompanied by inflammation in lamina propria. Furthermore correlation to clinical symptoms should be emphasized as histology is not specific.

A: Our diagnosis was based on the combination of increased number of IFLs, clinical symptoms and histological determinations. In light of the changes we have made in response to previous comments, we think that this problem is sufficiently addressed in the revised manuscript.

C: Line 237: Change “earlier” to “previous”. Line 238: Change “surprised” to “surprising”. Line 247: Maybe “secreted” is preferable to “broken”. Line 250: As line 238. Line 268, 269, 270: Change “LA” to “LC”. Line 270: Add “activity”, se comment in abstract section

A: We have followed all these suggestions.

Round 2

Reviewer 2 Report

The authors have significantly addressed most of the comments and suggestions for change that I had previously made.  However, there are lingering issues with syntax/grammar throughout the manuscript.  More importantly, though, they have confirmed that they did not accurately assess for enterochromaffin cells (i.e. by not incorporating a 5-HT specific antibody in their IHC approach).  Thus, part of their analysis and conclusions (at least those they attribute to changes in EC cells) can not be reliably supported.  My recommendations are as follows...

1) Work with an English-speaking editor to properly edit this manuscript and address numerous mistakes throughout.

2) If the authors are unable/unwilling to perform the appropriate immunohistochemical tests required to accurately evaluate for EC cells (i.e. by including 5-HT-specific staining), they need to adjust the manuscript to properly describe what they actually did in this regard.  In this case, they were assessing for enteroendocrine cells (NOT EC cells specifically).  This is a broader groups of cell types that includes EC cells, of course, but should not be considered a direct measure of the latter.  It still may be useful to report enteroendocrine cell prevalence in this context. However, I would not publish this manuscript as is, though, until they address this and #1.